# Predictive Factors of Self-Reported Quality of Life in Acquired Brain Injury: One-Year Follow-Up

**DOI:** 10.3390/ijerph18030927

**Published:** 2021-01-21

**Authors:** Alba Aza, Miguel Á. Verdugo, María Begoña Orgaz, Antonio M. Amor, María Fernández

**Affiliations:** 1Institute for Community Inclusion, Department of Personality, Assessment and Psychological Treatments, Faculty of Psychology, University of Salamanca, 37005 Salamanca, Spain; verdugo@usal.es (M.Á.V.); aamor@usal.es (A.M.A.); mariafernandez@usal.es (M.F.); 2Institute for Community Inclusion, Department of Basic Psychology, Psychobiology and Behavioral Sciences Methodology, Faculty of Psychology, University of Salamanca, 37005 Salamanca, Spain; borgaz@usal.es

**Keywords:** acquired brain injury, CAVIDACE scale, longitudinal study, predictors, quality of life, self-reported outcomes

## Abstract

*Background:* The sequelae and disabilities that follow an acquired brain injury (ABI) may negatively affect quality of life (QoL). The main objective of the study is to describe the QoL after an ABI and identify the predictors of a better QoL. *Methods*: Prospective cohort study with follow-up measurement after one-year. The sample comprised 203 adults with ABIs (64% male) aged 18–86 years (*M* = 53.01, *SD* = 14.44). Stroke was the main etiology of the injury (55.7%), followed by a TBI (32.8%), and the average time since injury was 8 years (*M* = 8.25, *SD* = 7.83, range = 0.5–47.5). Patients assessed their QoL through the scale *Calidad de Vida en Daño Cerebral* (CAVIDACE self-reported version; “quality of life in brain injury” in English), an ABI-specific tool based on the eight-domain QoL model. Other variables measured were: depression, self-awareness, community integration, resilience, and social support at baseline and one-year follow-up. *Results*: The studied factors showed few significant changes over time. The analyses showed statistically significant differences in QoL scores in several sociodemographic (age, civil status, education, legal capacity, and dependency), injury-related (time, location, and comorbidity), rehabilitation, and personal-social variables (self-awareness, depression, social support, resilience, and community integration). The levels of dependency, depression, and satisfaction with social support were independent predictors of the total QoL score one-year follow-up. *Conclusions*: QoL after ABI depends on multiple elements that must be considered. There are factors such as satisfaction with social support, depression, community integration, and resilience that must be monitored throughout the rehabilitation process.

## 1. Introduction

Acquired brain injuries (ABIs) are caused by a sudden injury in the brain that occurs after birth and includes different diagnoses such as traumatic brain injury (TBI), stroke, brain tumor, anoxia, and infection. In Spain, there is currently a prevalence of 420,064 people living with ABI, and there are approximately 104,701 new cases per year [1], imposing considerable costs on society due to the years of life lost to disability or death [2]. An ABI is often accompanied by long-lasting or permanent physical (i.e., spasticity, mobility problems, and chronic pain), cognitive (i.e., executive functioning, attention, memory and learning, communication, and anosognosia), emotional (i.e., anxiety and depression), and social impairments (i.e., social isolation and inability to return to work) [3,4,5,6,7,8] that negatively affect quality of life (QoL) [8,9,10,11,12].

QoL has been recognized as an important outcome of the rehabilitation process after a brain injury. QoL after an ABI has been discussed and conceptualized using the health-related QoL approach (HRQoL). This model focuses mainly on the impact of this medical condition and the rehabilitation process on physical, emotional, and social aspects, ignoring other aspects that are very important for personal well-being, such as self-determination, interpersonal relationships, and personal development. Therefore, we propose a comprehensive approach for QoL assessment based on Schalock and Verdugo’s QoL model [13,14], characterized by a broad range of personal outcomes. According to these authors, QoL is a multidimensional phenomenon that reflects the well-being desired by the person in relation to eight basic domains: emotional well-being, interpersonal relationships, material well-being, personal development, physical well-being, self-determination, social inclusion, and rights. These areas are assessed through culturally sensitive indicators and influenced by personal characteristics, environmental factors, and their interaction. The core domains are the same for all people, although they may vary individually in their relative value and importance [15,16]. In addition, from this model, it is possible to gather information from both self-reported and proxy perspectives. On one hand, the presence of communication and memory alterations [7,17,18] and a lack of self-awareness (i.e., anosognosia) [19,20] are very frequent among the population with ABIs and can affect the validity of self-reported scores. On the other hand, data from proxy reports should also be taken with caution since there is no sufficient correlation between proxy and self-reports [20].

Several studies have shown that the QoL after ABIs is worse than that in the general population [9,10,17,21,22,23,24,25]. However, QoL can change over time since the recovery process after an injury is long and complex. Improvement in QoL is generally experienced up to one [6,8,22,23,24,26] or two [12,25] years after an injury, and, afterward, the levels remain more or less stable [4,7,10]. Although this depends on the QoL area evaluated, findings show higher rates of change in the physical domain than in the emotional and social domains [12,24,25,26].

The course of evolution of the QoL after an ABI may vary due to different factors, such as sociodemographic, injury-related, personal, and social factors. Chief among the sociodemographic characteristics are gender, age, and employment status. Regarding gender or age, although some studies indicate that females [21,22,25,26] and elderly individuals [21,22,25,26] have a worse QoL, the results are not always clear [3,9,11,27,28]. Regarding employment, both employment before an ABI [3,10,29] and, even more so, an active employment situation after it [6,9,11,26,28,30,31,32] have been equally–and quite unequivocally related to a better QoL.

Regarding the published research on injury-related factors, it reveals conflicting results concerning the relationship between injury severity [3,21,22,24,30] or an ABI’s etiology [29,33] and QoL. However, there is an agreement that a great number of impairments after an ABI (i.e., comorbidity) are related to a worse QoL [3,17,22,31,34,35]. Most of the studies have focused their attention on the personal and social variables that could affect QoL, and that could be modified. Thus, we know that the absence of depressive symptoms [4,5,9,10,24,27,33,34,36], good social support [8,27,28,30,31], adequate community integration [4,10,23], and a resilient personality [37,38] have strong relationships with a better QoL. Other factors such as self-awareness have also been found to affect QoL, but the direction of this influence remains unclear [20,39].

This manuscript is based on extensive research about QoL after an ABI [40,41]. In another manuscript by the present authors [42], in which it was examined the changes in QoL between baseline and one-year follow-up considering both the assessments made by persons with ABI (i.e., self-reported) and that of their relatives and professionals (i.e., proxy-report), significant positive changes were found in the total QoL score and for nearly all the QoL domains (emotional well-being, material well-being, personal development, physical well-being, and rights). As for this study, the present authors’ focus is put on examining predictive factors of QoL over a one-year follow-up since the baseline, using a multidimensional model of QoL and self-report assessments. Specifically, we aimed at (1) describing changes in QoL after an ABI between baseline and one-year follow-up evaluation, (2) describing and analyzing the changes in important variables (i.e., depression, self-awareness, community integration, resilience, and social support) at one-year follow-up, and (3) examining the impact of sociodemographic, injury-related, personal, and social variables on QoL and identify the predictors of a better QoL.

## 2. Materials and Methods

### 2.1. Study Participants

A prospective one-year follow-up was conducted with a cohort of adults with ABIs from 26 rehabilitation centers that provide health and social services in Spain. The ABI participants had to meet the following inclusion criteria: (a) to have an ABI, (b) to be at least 16 years, (c) to attend an ABI-specific center, and (d) to have signed an informed consent form. The exclusion criteria were the following: (a) to be in a state of coma or having minimum consciousness, (b) to suffer from global aphasia, and (c) not to be able to understand or answer most questions. The individuals with ABIs responded to a QoL measure and a series of complementary questionnaires. Of the 402 participants in the baseline, 199 (49.5%) dropped out at the one-year follow-up due to the end of their rehabilitation, death, or refusal to continue in the study.

### 2.2. Measures

#### 2.2.1. CAVIDACE Scale

The scale *Calidad de Vida en Daño Cerebral* (CAVIDACE; “quality of life in brain injury” in English) has been specifically designed to measure the QoL of adults with ABIs using proxy responses [43]. We used the self-reported version: an adaptation of the original scale completed by individuals with ABIs. This version consisted of 40 items, which assessed the eight domains that are subsumed by Schalock and Verdugo’s model: emotional well-being, interpersonal relationships, material well-being, personal development, physical well-being, self-determination, social inclusion, and rights. The responses were recorded on a four-point rating scale: 0 = never, 1 = sometimes, 2 = frequently, and 3 = always. The instrument includes negatively worded items, which were reversed prior to adding the scores of the items per each domain. These direct scores are transformed into standard scores for each domain (*M* = 10, *SD* = 3) and percentiles. Moreover, the scale provides an over-all raw QoL score (i.e., the sum of the direct scores obtained in each of the domains) that may vary from 0 to 120, where higher scores indicate higher QoL. This overall score can be converted into an easily interpretable QoL Global Index (*M* = 100; *SD* = 15). Its psychometric properties were good and comparable to those of the original scale: QoL is composed of eight first-order intercorrelated domains (CFI = 0.891, RMSEA = 0.050, TLI = 0.881), and the internal consistency was adequate in seven of the eight domains (*ω* = 0.66–0.87) [40].

#### 2.2.2. Patient Health Questionnaire-9 (PHQ9)

The PHQ9 [44], which consists of 9 items, assessed depression in accordance with the DSM-IV criteria. Total scores can range from 0 to 27, and higher scores are indicative of severe depression.

#### 2.2.3. Patient Competency Rating Scale (PCRS)

The PCRS consists of 30 items that assessed the competency to perform different daily living tasks [45]. To complete the instrument, individuals with ABI were required to indicate the extent to which it was difficult for them to perform the task that was described in each item. Participant responses can be compared with those of a family member or professional to determine the self-awareness level. In this study, we compared the responses of individuals with ABI and professionals. The wider the discrepancies found between individuals with ABI and professionals, the poorer the self-awareness.

#### 2.2.4. Community Integration Questionnaire (CIQ)

The CIQ is a 15-item specific measure of community integration [46]. Total scores can range from 0 to 29. We used the version of Sander et al. [47] in which ambiguous items were eliminated (i.e., 4 and 10).

#### 2.2.5. Connor–Davidson Resilience Scale (CD-RISC)

The CD-RISC is a 25-item measure of resilience [48] that has been used in samples with a wide range of conditions, including ABIs (scores range from 0 to 100).

#### 2.2.6. Social Support Questionnaire-6 (SSQ6)

The SSQ6 is an abbreviated version of the social support questionnaire [49]. Individuals were required to respond to the 6 items by (a) indicating the number of individuals available to support them and (b) rating their level of satisfaction with social support. Scores can range from 0 (no social support) to 6 (very high social support) for the number of available supports, and from 1 (very unsatisfied) to 6 (very satisfied) for the satisfaction domain in each item or area. From these scores in the 6 areas, an average score was calculated for the number of available supports and for satisfaction.

### 2.3. Procedure

Participating organizations that provide attention to the ABI population were recruited through emails and telephone calls. First, we contacted the centers that had participated in the study in which the CAVIDACE scale was developed and validated. On numerous occasions, these professionals facilitated liaisons with other centers, thereby resulting in snowball sampling. Second, to recruit a larger sample, information about the study was disseminated through conferences and posted on the university’s website. Of the 32 centers with which the research team made initial contact, 26 finally agreed to participate in the study.

Participants attended ABI-specific care centers (i.e., rehabilitation centers and day centers) spread throughout Spain. These are socio-sanitary centers focused mainly on a chronic phase of ABI (although when the rehabilitation phase within hospitals is brief due to time or resources constrain, people with ABI in the subacute phase can also be sent to these centers). The main difference between rehabilitation centers and daycare centers is the purpose of the services they provide: daycare centers provide daily care whose aim is to improve or maintain personal autonomy to an adequate level and to provide families with support to alleviate the burden caused by the ongoing support they commonly give (i.e., family respite service). The rehabilitation centers, for their part, pursue a therapeutic objective aimed at re-educating and compensating for the consequences of the injury, preventing future complications, and improving the preserved abilities

Once a center expressed interest in participating, a research team member visited it and provided all necessary information about the study. In each center, a research assistant was trained to oversee the administration of the instruments. To participate in the study, participants had to meet the established inclusion criteria, and when the number of participants exceeded the possibilities of participation of the center, it was the research assistant who randomly selected participants. In addition, the research assistant (in consensus with a professional from the center, when necessary) was in charge of determining the ability of the person with ABI to answer the instruments. The research team provided printed copies, although respondents were able to complete the scales online as well.

A follow-up was carried out after one year, and the follow-up used the same instruments. Demographic and clinical information was only provided at the baseline. Finally, the scales were collected, and the data were analyzed.

All the procedures described in this paper followed the ethical standards required by research that involves human participants. This study was approved by the bioethics committee of the University of Salamanca (No: 20189990014185/ Record: 2018/REGSAL-1931). Written informed consent was obtained from the ABI participants. Personal and clinical data were collected, stored, and protected in accordance with the Organic Law 3/2018 of 5 December on Data Protection and the Guarantee of Digital Rights, so alphanumeric codes were assigned to all the participants to guarantee their anonymity. All procedures comply with the principles of the 1964 Declaration of Helsinki and its amendments.

### 2.4. Statistical Analysis

Data were analyzed using SPSS 24, and statistical significance was set at *p* = 0.05. Descriptive data are displayed as the mean, *SD,* and range or absolute and relative frequencies. When comparing characteristics between the patients included and those who were lost for the follow-up, the categorical variables were analyzed with a chi-squared test and the continuous variables with an independent *t*-test.

To verify the effect that the time elapsed since ABI could have on the changes experienced in the QoL, the sample was divided between those who had recently had the ABI (i.e., 3 years ago or less) and those who were in a chronic phase (i.e., those who had the ABI more than 3 years ago) and carried out repeated measured *t*-test between QoL at baseline and one-year follow-up for QoL’s domains and total QoL index.

Paired t-tests were used to compare PHQ9, PCRS, CIQ, CD-RISC, and SSQ6 from the baseline to the one-year follow-up. Confidence interval plots were used to represent the results.

Analysis to determine the predictors of the QoL scores was conducted. Before carrying out the analyses, we implemented a transformation of the quantitative scales (i.e., PHQ9, PCRS, CIQ, CD-RISC, and SSQ6) in categories (i.e., low, intermediate, and high) from the calculation of the percentiles. First, comparisons between groups were performed using independent-sample *t*-tests and ANOVA with Tukey *post-hoc* tests. The effect size was analyzed using eta-squared (*η^2^*). Second, to identify the prediction of the dependent variables (i.e., the QoL total score and domains at the one-year follow-up), hierarchical multiple linear regression models were conducted with the following six groups of independent variables: Step 1 (QoL scores at the baseline), Step 2 (sociodemographic variables), Step 3 (injury-related variables), Step 4 (the type of rehabilitation center), Step 5 (personal and social variables at the baseline), and Step 6 (personal and social variables at the one-year follow-up). First, the variables of each group that were significant in the previous analysis were included by step as an initial model (the enter method). Factors with *p* < *0*.10 were retained. Once the potential predictors were identified, the model was built via the forward method. All the variables fulfilled the assumptions of normality and no multicollinearity (*rs* < 0.70). The results are presented as adjusted *R^2^s*, F changes, and standardized betas. *R^2^* was interpreted according to Cohen’s [50] guidelines (i.e., 0.02 = small, 0.13 = medium, and 0.26 = large).

## 3. Results

### 3.1. Patient Sample

More than half of the participants were male (64%) aged from 18 to 86 years (M = 53.01, SD = 14.44). There was a low percentage of subjects who returned to work or their studies after their injury (2%) and a low frequency of people living independently (9.8%). Stroke was the main etiology of the injury (55.7%), followed by a TBI (32.8%), and the average time since injury was 8 years (M = 8.25, SD = 7.83, range = 0.5–47.5). See Table 1 for more information. When comparing the differences between patients with and without follow-ups, significant differences were found in time since injury (t_371_ = −10.95, *p* < 0.01), etiology (χ² = 24.30, *p* < 0.001), and type of center (χ² = 23.36, *p* < 0.001). People for whom one-year follow-up evaluations were not conducted had their ABI more recently, went more to rehabilitation centers than to daycare centers, and had a higher prevalence of stroke and a lower prevalence of a TBI.

At the one-year follow-up, the average score in the Total QoL Index was 105.11 (SD = 15.51). By domains, we found the highest value in material well-being and rights and the lowest values in social inclusion and personal development. These descriptive results are listed in Table 2.

### 3.2. Changes in QoL from Baseline to One-Year Follow-Up

Improvements in QoL levels have been reported between baseline and one-year follow-up (see Table 3 for more information). However, these changes have only been significant in emotional well-being for the group with the most recent ABI (i.e., ABI 3 years ago or less) and in personal development for the chronic group (i.e., ABI more than 3 years ago).

### 3.3. Changes in Other Variables from the Baseline to One-Year Follow-Up

Statistically significant differences between baseline and one-year follow-up were found only for PHQ9 (t_140_ = 2.10, *p* = 0.038) and CD-RISC (t_129_ = −2.02, *p* =0.045), showing a decrease in depression over time (M_baseline_ = 7.15, SD_baseline_ = 5.72; M_follow-up_ = 6.16, SD_follow-up_ = 5.96) and an improvement in resilience (M_baseline_ = 62.79, SD_baseline_ = 17.52; M_follow-up_ = 65.89, SD_follow-up_ = 18.61). These results are represented in Figure 1.

### 3.4. Factors Related to QoL: Independent t-Tests and ANOVAs

Statistically significant differences were found in the total QoL index scored by loss of legal capacity, comorbidity, depression, resilience, and satisfaction with social support. No significant differences were detected in any of the QoL scores (*p* > 0.05) according to gender, prior employment, level of support, time since injury, etiology, self-awareness (baseline and one-year), and social support (one-year). Table 4 shows the results of the analysis. Higher levels of QoL were related to people who were younger, were married or with a partner, had higher education, were legally capable, had a lower degree of dependency, had a unilateral ABI, had fewer associated health conditions (i.e., comorbidity), were receiving care in a rehabilitation center, had lower levels of depression, and had higher levels of social support, resilience, and community integration.

### 3.5. Predictors of QoL at One-Year: Regressions

Nine hierarchical multiple regressions were conducted to examine the potential predictors of ABI participants through the analysis of the variance in QoL. According to Cohen’s guidelines, we found large predictive values in all cases except for material well-being. However, if we neglect the effect of levels of QoL in the baseline, we found that the best models were in the emotional well-being, personal development, and rights variables.

If we focus on that model, we found that personal and social variables were present in all the explanatory models. Specifically, depression_baseline_ was an explanatory factor of emotional well-being (*B* = −0.33, *p* = 0.002), resilience_12 months_ of emotional well-being and personal development (*B* = 0.34, *p* = 0.002; *B* = 0.45, *p* < 0.001, respectively), satisfaction with social support_baseline_ predicted rights (*B* = 0.47, *p* = 0.001), and community integration_baseline_ predicted personal development and rights (*B* = 0.25, *p* = 0.004; *B* = 0.28, *p* = 0.041, respectively). Furthermore, marital status (i.e., a sociodemographic variable) was present in the explanatory model of emotional well-being (*B* = 0.24, *p* = 0.020) and the loss of legal capacity (i.e., injury-related variable) in personal development (*B* = −0.16, *p* = 0.043). See Table 5 for more information.

## 4. Discussion

This study presents the QoL outcomes, as measured by the CAVIDACE scale, at the one-year follow-up in a sample of ABI adults who had experienced ABI some years ago. Likewise, it also explores the changes at the one-year follow-up of some personal and social variables and the association between QoL and these factors and a set of sociodemographic and injury-related factors that were expected to predict the patient’s QoL. There are many studies that analyze how some of these factors affect QoL; however, they usually focus only on a few variables, use cross-sectional designs, or follow an HRQoL model. In this sense, the study contributes by bettering the knowledge and understanding of the QoL construct in the ABI population.

At the one-year follow-up, the domains with the highest results were rights and material well-being, whereas personal development obtained the lowest scores. These results are consistent with those obtained in other studies of those with ABIs [29,41] and studies assessing the recipients of social services [51], suggesting the importance of promoting community integration and cognitive skills in the population with ABIs. The findings of this study show that improvements in QoL were generally experienced between the baseline evaluation and the one-year follow-up, but they were only significant for the emotional well-being and personal development domains. In other studies [42] that analyzed different change patterns as a function of the time elapsed since the ABI, more significant changes were obtained when ABI was recent; however, this finding has not been replicated in the present study. The determining factor for such difference may be that in this case, self-report assessments were analyzed and not proxy assessment, which was used in other studies.

Social support, depression, self-awareness, community integration, and resilience are important aspects that have been widely studied in the population with ABIs [5,10,28,36,39]. At the one-year follow-up, an improvement was found in SSQ6, CIQ, and CD-RIS outcomes, although only the last one was significant. Meanwhile, PHQ9 scores decreased significantly, and the PCRS scores remained stable. According to the literature, there are studies that have reported changes in satisfaction with social support, which are not necessarily positive; and no changes in the number of supports [30]. Studies about community integration indicate the most important changes are one-year after an ABI and small improvements later [52], which agrees with the results of this study. This suggests the need to further improve support and social interaction, particularly at the community level, including patient organizations and support groups [53,54] and the promotion of self-determined and active work life. The few available studies about resilience found a stabilization period of one year after an ABI and a subsequent worsening [55]. Finally, there is no consensus about depression’s evolutionary patterns. While some studies report a higher prevalence with more time elapsed since an ABI [10], others report improvements [7,56]. For all the variables, we must consider that longer time periods may be needed to appreciate any changes and the importance of analyzing the moderating variables of these results.

Past findings have yielded contradictions about the effects of sociodemographic variables on QoL. In this study, no significant effects were found for gender, as happened in others [10,57], or for employment before an ABI. Those people with ABIs who were working or studying before the injury showed significantly better QoL in previous research [29,58], as in our baseline evaluation [41], which may indicate that the positive effect of a previous active lifestyle disappears over time. In addition, people who were married or cohabitating reported better emotional well-being, reflecting the importance of close interpersonal relationships as a preventive factor for depression and anxiety [59,60], although there is no unanimity in this regard [58]. Those with higher levels of education presented better levels of self-determination, possibly related to greater possibilities of acquiring the work and lifestyle desired [5,58].

Regarding the injury-related variables, it was found that people with a deprived legal capacity showed a lower QoL, probably due to the importance of being able to make preference-based elections/decisions/choices. In addition, the higher the level of recognized dependency (i.e., the need for support or supervision to perform daily life activities), the poorer the levels of QoL [28,29,61]. Decision-making abilities, the capacity to act independently, and participation in inclusive settings tended to be poorer among individuals who had a high dependency. Finally, a better QoL was also found in people who had unilateral injuries compared to those with bilateral injuries, probably due to a lower number of associated sequelae [41]. In this sense, lower comorbidity levels at the baseline were significantly related to a better QoL at the one-year follow-up [22,35,62]. Neither the etiology [58] nor the time elapsed since the injury [28,34] had a significant impact on QoL.

Depression, satisfaction with social support, community integration, and resilience were predictors of self-reported QoL at the baseline and one-year follow-up. Depression was a predictor of emotional well-being and the total QoL index. Depressive problems a year ago (i.e., baseline evaluation) continued to affect the ABI person’s emotional well-being and had a negative effect on other aspects of QoL at the time, which had already been documented by other studies [9,33,56]. The effect of community integration on QoL has been widely reported, but the effect was not found for resilience. However, both have been shown to be the two most important predictors. Community integration seems to exert its fundamental effect based on the levels found in the initial evaluation, contrary to what happens with resilience, and its broad and lasting effects make it difficult to produce improvements over time [52]. The exceptions are the effect of community integration at the present time on material well-being and interpersonal relationships, which could be closely related to productivity levels, salary, and the availability of current contexts suitable for establishing interpersonal relationships. In the case of resilience, the immediate effects it has on QoL are consistent with the importance of deploying coping skills focused on the present moment and not on evaluations of what the future will be like.

Finally, satisfaction with social support [28] had an immediate effect on interpersonal relationships and physical well-being and a long-term effect on rights and the total QoL index. It is possible that the existence of quality social relationships ensures better support for one’s physical needs associated at the time and that the existence of these supports allows the introduction of progressive changes that ensure respect for the rights of the ABI person. It should be noted that the number of social supports and self-awareness variables did not have significant effects on self-reported QoL. However, it was demonstrated that the sample had a lack of self-awareness, and its effect was possibly reflected in the evaluations carried out by others [41].

Some study limitations need to be addressed. The sample of the respondents was recruited using a non-probabilistic convenience sampling process, which limits the generalizability of the findings. Second, no objective test was used to determine the ability of people with ABI to participate in the study. This could introduce some subjectivity, even though the research team strictly controlled it and the professionals acted on their clinical judgment based on their knowledge of the individuals with ABIs and their clinical history. Finally, there was a high percentage of experimental death, even though it was a relatively short follow-up. However, it was shown that there are no substantial differences between those who dropped out of the study and those who continued.

## 5. Conclusions

We have shown that satisfaction with social support, depression, community integration, and resilience are the main predictors of self-reported QoL levels in patients with ABIs. However, hardly any changes were found in these variables over the course of a year. This implies the need to implement a greater number of programs and early actions on these aspects for clinical practice, being especially important the prevention and detection programs. Furthermore, these results highlight the importance of psychological, neuropsychological, and occupational therapy interventions as a part of the care of ABI, which was limited in many cases to physical aspects. Future lines of work should include broader longitudinal follow-ups, as well as analysis of the specific effects that some of the interventions carried out in this population have on QoL.

## Figures and Tables

**Figure 1 ijerph-18-00927-f001:**
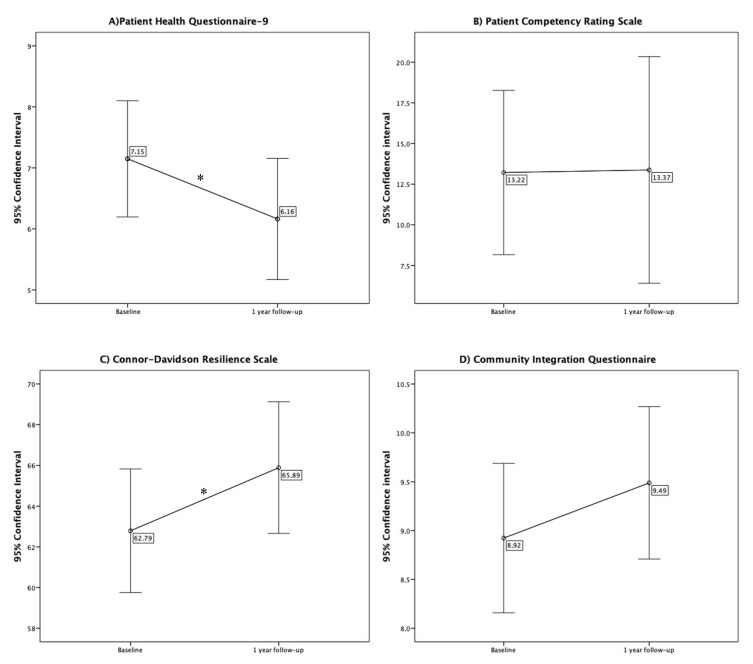
Confidence interval plots for the scores in (**A**) Patient Health Questionnaire-9 (PHQ9), (**B**) Patient Competency Rating Scale (PCRS), (**C**) Connor–Davidson Resilience Scale (CD-RISC), (**D**) Community Integration Questionnaire (CIQ), and (**E**) Social Support Questionnaire-6 (SSQ6) in baseline and 1 year follow-up (Note: * *p* < 0.05 in paired *t*-test analysis).

**Table 1 ijerph-18-00927-t001:** Sociodemographic and clinical characteristics of the acquired brain injuries (ABIs) sample.

Sociodemographic and Clinical Variables	Patients with 12 Months Follow-Up (*n* = 203)	Patients without Complete Follow-Up (*n* = 402)	
	*n* (%)	*n* (%)	
*Gender*	203	400	*χ*² = 1.87
Male	130 (64%)	243 (60.8%)	
Female	73 (36%)	157 (39.3%)	
*Age (years)*	203	396	*t_394_* = −0.12
Mean (SD)	53.01 (14.44)	54.83 (14.47)	
Range	18–86	18–91	
*Civil status*	198	391	*χ*² = 2.46
Single/separated/divorced/window(er)	107 (54%)	196 (50.1%)	
Married/cohabitating	91 (46%)	195 (49.9%)	
*Educational level*	190	369	*χ*² = 1.02
Without education/none	14 (7.4%)	30 (8.1%)	
Primary education	60 (31.6%)	116 (31.4%)	
Secondary education	64 (33.7%)	117 (31.7%)	
Higher education	52 (27.4%)	106 (28.7%)	
*Prior employment status*	200	383	*χ*² = 2.58
Not active/unemployed	63 (31.5%)	135 (25.2%)	
Employed/student	137 (68,5%)	248 (64.8%)	
*Current employment status*	203	386	*χ*² = 0.65
Not active/unemployed	199 (98%)	376 (97.4%)	
Employed/student	4 (2%)	10 (2.6%)	
*Type of home*	122	251	*χ*² = 0.90
Independent flat	12 (9.8%)	29 (11.6%)	
Residential center	18 (14.8%)	39 (15.5%)	
Family home/sheltered flat	92 (75.4%)	183 (72.9%)	
*Level support*	189	355	*χ*² = 9.24 *
Intermittent	20 (10.6%)	42 (11.6%)	
Limited	12 (6.3%)	38 (10.5%)	
Extensive	50 (26.5%)	98 (27.0%)	
Generalized	107 (56.6%)	185 (51.0%)	
*Loss of legal capacity*	191	369	*χ*² = 3.91
No	127 (66.5%)	262 (71%)	
Yes	64 (33.5%)	107 (29%)	
*Dependence recognized*	194	370	*χ*² = 0.00
No	44 (22.7%)	84 (22.7%)	
Yes	150 (77.3%)	286 (77.3%)	
*Degree of dependence*	153	276	*χ*² = 1.97
Grade I moderate dependency	24 (15.2%)	40 (14.5%)	
Grade II severe dependency	51 (33.3%)	102 (37.0%)	
Grade III major dependency	78 (51%)	134 (48.6%)	
*Time since the injury (years)*	194	373	*t_371_* = 10.95 **
Mean (SD)	8.25 (7.83)	7.20 (6.98)	
Range	0.5–47.5	0.5–47.5	
*Location of the injury*	191	358	*χ*² = 4.90 *
One hemisphere	121 (63.4%)	245 (68.4%)	
Both hemispheres	70 (36.6%)	113 (31.6%)	
*Etiology of the injury*	201	390	*χ*² = 24.30 ***
Stroke	112 (55.7%)	239 (61.3%)	
Traumatic brain injury	66 (32.8%)	93 (23.8%)	
Cerebral anoxia	10 (5%)	16 (4.1%)	
Cerebral tumors	6 (3%)	17 (4.4%)	
Infection diseases	2 (1%)	8 (2.1%)	
Other	5 (2.5%)	17 (4.4%)	
*Comorbidity (health conditions)*	203	393	*t_400_* = −2.12
Mean (SD)	5.35 (2.49)	5.01 (2.44)	
Range	0–12	0–12	
*Type of center*	171	313	*χ*² = 23.36 ***
Day center	101 (59.1%)	146 (46.6%)	
Rehabilitation center	70 (40.9%)	167 (53.4%)	

Note: * *p* < 0.05, ** *p* < 0.01, *** *p* < 0.001.

**Table 2 ijerph-18-00927-t002:** Description statistics of quality of life (QoL) scores one-year follow-up.

Statistics	EW	IR	MW	PD	PW	SD	SI	RI	Total QoL Index
No. items	5	5	5	5	5	5	5	5	40
*Mean*	11.23	10.25	12.33	9.55	10.93	10.26	9.27	12.06	105.11
*SD*	2.83	3.65	3.36	3.39	3.10	3.75	3.65	2.90	15.51
Range of scores	1–15	0–15	0–15	0–15	0–15	0–15	0–15	3–15	71–135
Skewness	−0.71	−0.35	−1.57	−0.37	−0.38	−0.46	−0.20	−0.94	0.07
Kurtosis	0.26	−0.68	2.26	−0.44	−0.32	−0.75	−0.42	0.22	−0.60

Note. EW = emotional well-being; IR = interpersonal relationships; MW = material well-being; PD = personal development; PW = physical well-being; SD = self-determination; SI = social inclusion; RI = rights.

**Table 3 ijerph-18-00927-t003:** Results by groups of time since the injury of repeated measured *t*-test between baseline and one-year follow-up in QoL’s domains and total score.

Domain	ABI 3 Years Ago or Less (*n* = 63)				ABI More Than 3 Years Ago (*n* = 131)			
	*t*	Effect Size	Baseline	One- Year Follow-Up	*t*	Effect size	Baseline	One- Year Follow-Up
EW	*t_(57)_* = −9.42 **	0.14	10.48 (3.29)	11.72 (2.88)	*t_(95)_* = −0.05	0.00	10.98 (2.84)	11.01 (2.81)
IR	*t_(54)_* = 0.46	0.01	11.29 (3.09)	10.96 (3.31)	*t_(97)_* = −1.17	0.01	9.55 (3.68)	9.92 (3.63)
MW	*t_(56)_* = −0.07	0.00	12.28 (2.80)	12.40 (3.31)	*t_(96)_* = −2.82	0.03	11.88 (3.05)	12.38 (3.37)
PD	*t_(57)_* = −1.35	0.02	9.22 (3.52)	9.81 (3.65)	*t_(98)_* = −3.86 *	0.04	8.82 (2.98)	9.39 (3.31)
PW	*t_(57)_* = −0.01	0.00	11.31 (2.93)	11.36 (3.21)	*t_(92)_* = 0.00	0.00	10.61 (2.81)	10.60 (3.10)
SD	*t_(55)_* = 0.04	0.00	11.02 (3.01)	10.91 (3.97)	*t_(95)_* = −1.79	0.02	9.38 (3.89)	9.87 (3.60)
SI	*t_(57)_* = −0.51	0.01	8.74 (4.04)	9.19 (3.65)	*t_(97)_* = −1.64	0.02	8.93 (3.97)	9.40 (3.51)
RI	*t_(55)_* = 0.29	0.00	12.64 (2.49)	12.39 (3.42)	*t_(95)_* = −0.01	0.00	11.85 (2.76)	11.88 (2.63)
Total	*t_(33)_* = 0.03	0.00	103.44 (14.14)	102.91 (14.65)	*t_(95)_* = −1.79	0.03	101.91 (14.73)	103.80 (15.48)

Note: data are presented as mean and standard deviation (SD), EW = emotional well-being; IR = interpersonal relationships; MW = material well-being; PD = personal development; PW = physical well-being; SD = self-determination; SI = social inclusion; RI = rights, * *p* < 0.05, ** *p* < 0.01.

**Table 4 ijerph-18-00927-t004:** Parametric test results for QoL’s domains and total score one-year follow-up.

	EW	IR	MW	PD	PW	SD	SI	RI	Total QoL Index
Sociodemographic Variables									
*Gender*									
*Age* 0 = ≤50 1 = >50						*η*^2^ = 0.03 * 10.96 (3.25) 9.74 (4.02)			
*Civil status* 0 = Single/separated/divorced 1 = Married/cohabitating	*η*^2^ = 0.03 * 10.76 (2.77) 11.69 (2.84)								
*Educational level* 1 = Without education/none 2 = Primary education 3 = Secondary education 4 = Higher education						*η*^2^ = 0.05 * 8.17 (3.41) 9.57 (3.89) 10.41 (3.48) 11.23 (3.88)			
*Prior employment*									
*Type of home* 1 = Independent flat 2 = Residential center 3 = Family/sheltered			*η*^2^ = 0.15 *** 12.82 (1.89) 8.87 (3.91) 12.66 (3.63)					*η*^2^ = 0.08 * 12.64 (1.69) 9.93 (3.81) 12.24 (2.85)	
**Injury-related variables**									
*Level of support*									
*Loss of legal capacity* 0 = No 1 = Yes		*η*^2^ = 0.03 * 10.69 (3.40) 9.26 (4.02)		*η*^2^ = 0.06 ** 10.11 (3.29) 8.28 (3.44)					
*Degree of dependency* 1 = Grade I moderate dependency 2 = Grade II severe dependency 3 = Grade I major dependency						*η*^2^ = 0.02 * 11.81 (3.03) 9.29 (3.69) 9.23 (3.85)			*η*^2^ = 0.02 ** 116.28 (14.34) 102.58 (14.82) 100.68 (15.15)
*Time since injury*									
*Location of the injury* 0 = Unilateral 1 = Bilateral		*η*^2^ = 0.03 * 10.71 (3.56) 9.54 (3.44)		*η*^2^ = 0.04 * 9.98 (3.46) 8.63 (3.12)			*η*^2^ = 0.03 * 9.82 (3.63) 8.47 (3.27)		
*Etiology of the injury*									
*Comorbidity* 0 = ≤5 1 = >5	*η*^2^ = 0.04 ** 11.70 (2.55) 10.52 (3.09)			*η*^2^ = 0.04 ** 10.14 (3.18) 8.69 (3.54)		*η*^2^ = 0.04 * 10.83 (3.63) 9.40 (3.79)	*η*^2^ = 0.04 ** 9.89 (3.67) 8.37 (3.45)		*η*^2^ = 0.04 * 107.79 (15.00) 101.77 (15.61)
**Rehabilitation variable**									
*Type of center* 0 = Day center 1 = Rehabilitation center		*η*^2^ = 0.05 ** 9.49 (3.93) 11.12 (3.22)							
**Personal and social variables (baseline)**									
*Self-awareness*									
*Depression* 1 = Low 2 = Intermediate 3 = High	*η*^2^ = 0.14 *** 12.13 (2.59) 11.56 (2.27) 9.18 (3.33)			*η*^2^ = 0.06 ** 10.34 (3.37) 9.54 (3.21) 7.85 (3.70)					*η*^2^ = 0.06 * 107.23 (15.12) 106.48 (17.80) 96.95 (13.32)
*Resilience* 1 = Low 2 = Intermediate 3 = High	*η*^2^ = 0.06 * 10.40 (2.57) 11.14 (2.79) 12.25 (2.62)	*η*^2^ = 0.07 ** 8.63 (4.02) 10.34 (3.21) 11.27 (3.77)	*η*^2^ = 0.04 * 11.09 (3.97) 12.69 (2.67) 12.62 (3.74)	*η*^2^ = 0.05 * 8.15 (3.29) 9.74 (3.19) 10.14 (3.55)					*η*^2^ = 0.12 *** 94.81 (12.00) 105.87 (13.05) 110.03 (18.93)
*Social support* 1 = Low 2 = Intermediate 3 = High		*η*^2^ = 0.05 * 9,45 (4.32) 10.30 (3.50) 11.54 (2.96)						*η*^2^ = 0.05 * 11.21 (3.27) 11.91 (3.08) 12.92 (2.17)	
*Satisfaction with social support* 1 = Low 2 = Intermediate 3 = High	*η*^2^ = 0.10 ** 9.64 (2.43) 11.13 (2.78) 12.08 (2.79)	*η*^2^ = 0.08 ** 8.92 (3.31) 9.48 (3.61) 11.31 (3.33)	*η*^2^ = 0.08 ** 10.44 (4.24) 12.17 (2.88) 13.08 (3.06)		*η*^2^ = 0.06 * 9.54 (3.01) 10.58 (2.97) 11.65 (3.16)		*η*^2^ = 0.05 * 7.81 (3.42) 8.92 (3.67) 9.95 (3.74)	*η*^2^ = 0.15 *** 10.35 (3.06) 11.51 (2.97) 13.25 (2.42)	*η*^2^ = 0.09 ** 95.61 (14.72) 104.36 (16.63) 109.02 (13.65)
*Community integration* 1 = Low 2 = Intermediate 3 = High		*η*^2^ = 0.09 ** 8.50 (4.20) 10.49 (3.08) 11.32 (3.23)	*η*^2^ = 0.06 ** 11.20 (3.96) 12.58 (3.26) 13.20 (2.31)	*η*^2^ = 0.13 *** 8.14 (3.37) 9.26 (3.26) 11.24 (2.43)		*η*^2^ = 0.14 *** 8.89 (3.66) 9.83 (3.84) 12.36 (2.70)	*η*^2^ = 0.14 *** 8.89 (3.66) 9.83 (3.84) 12.36 (2.70)	*η*^2^ = 0.13 *** 7.34 (3.71) 9.47 (3.39) 10.80 (3.11)	
**Personal and social variables (one-year)**									
*Self-awareness*									
*Depression* 1 = Low 2 = Intermediate 3 = High	*η*^2^ = 0.17 *** 12.16 (2.50) 11.24 (2.66) 8.65 (3.01)	*η*^2^ = 0.08 ** 11.20 (3.66) 10.00 (3.93) 8.15 (3.23)							*η*^2^ = 0.08 * 106.00 (16.78) 108.37 (14.40) 97.15 (9.60)
*Resilience* 1 = Low 2 = Intermediate 3 = High	*η*^2^ = 0.11 ** 10.14 (2.87) 10.82 (2.90) 12.62 (2.44)		*η*^2^ = 0.08 * 10.59 (4.00) 12.30 (3.06) 13.27 (2.99)	*η*^2^ = 0.28 *** 6.90 (3.49) 9.13 (2.87) 12.00 (2.60)		*η*^2^ = 0.11 ** 8.14 (3.97) 9.97 (3.37) 11.69 (3.28)	*η*^2^ = 0.17 *** 7.33 (3.10) 9.27 (3.06) 11.53 (3.70)	*η*^2^ = 0.05 * 11.19 (3.08) 11.73 (2.89) 13.03 (3.06)	*η*^2^ = 0.10 * 96.29 (12.28) 106.10 (14.09) 110.71 (17.17)
*Social support*									
*Satisfaction with social support* 1 = Low 2 = Intermediate 3 = High	*η*^2^ = 0.06 * 10.07 (3.13) 10.88 (2.70) 11.98 (2.73)	*η*^2^ = 0.15 *** 6.88 (3.56) 10.03 (3.91) 11.30 (3.44)			*η*^2^ = 0.14 ** 9.81 (2.97) 9.80 (3.19) 12.16 (2.75)			*η*^2^ = 0.16 *** 9.25 (3.32) 11.85 (2.85) 12.79 (2.47)	*η*^2^ = 0.08 * 96.46 (16.27) 106.42 (14.79) 108.72 (14.12)
*Community integration* 1 = Low 2 = Intermediate 3 = High		*η*^2^ = 0.13 *** 8.47 (3.96) 10.35 (3.82) 11.76 (2.63)	*η*^2^ = 0.08 * 11.11 (4.17) 12.68 (3.27) 13.30 (2.04)	*η*^2^ = 0.17 *** 8.06 (3.42) 8.73 (3.04) 11.16 (2.74)		*η*^2^ = 0.11 ** 8.63 (3.32) 10.35 (3.46) 11.51 (3.57)	*η*^2^ = 0.07 * 8.38 (4.14) 9.22 (3.38) 10.57 (3.02)	*η*^2^ = 0.07 * 10.82 (3.49) 11.56 (3.05) 12.77 (2.47)	

Note. EW = emotional well-being; IR = interpersonal relationships; MW = material well-being; PD = personal development; PW = physical well-being; SD = self-determination; SI = social inclusion; RI = rights; Data are presented as mean and *SD*; QoL values and eta-squared (*η*^2^) values with significant results of independent *t*-test or ANOVA; * *p* < 0.05, ** *p* < 0.001, *** *p* < 0.0001.

**Table 5 ijerph-18-00927-t005:** Results from the hierarchical regressions models of QoL domains and total score.

Dependent Variables	Variables (Final Model)	Unstandardized Coefficients		Standardized Coefficients	95 CI Lower/Upper Bound	*t*	*p*	*F* Change	Change Adjusted *R^2^*
		*B*	*S.E.*	Beta					
EW	Civil status	1.31	0.55	0.24	0.21/2.42	2.38	0.020	4.97 *	0.05
	Depression baseline	−1.27	0.39	−0.33	−2.05/−0.49	−0.33	0.002	16.06 ***	0.17
	Resilience 12 m	1.34	0.41	0.34	0.52/2.17	3.26	0.002	10.63 **	0.10
								Total	0.32
IR	IR baseline	0.47	0.09	0.48	0.30/0.64	5.57	<0.001	55.14 ***	0.38
	Satisfaction with social support 12 m	1.14	0.42	0.23	0.31/1.98	2.72	0.008	9.39 **	0.06
	Community integration 12 m	1.00	0.38	0.38	0.24/1.77	2.62	0.011	6.85 *	0.04
								Total	0.48
MW	Community integration 12 m	1.67	0.64	0.36	0.38/2.96	2.60	0.012	6.77 *	0.11
								Total	0.11
PD	Loss of legal capacity	−1.19	0.58	−0.16	−2.35/−0.04	−2.05	0.043	7.51 **	0.06
	Community integration baseline	1.08	0.36	0.25	0.36/1.79	3.00	0.004	16.83 ***	0.13
	Resilience 12 m	2.29	0.42	0.45	1.46/3.12	5.47	<0.001	29.94 ***	0.18
								Total	0.37
PW	PW baseline	0.47	0.10	0.43	0.27/0.67	4.69	<0.001	28.55 ***	0.23
	Satisfaction with social support 12 m	0.98	0.40	0.23	0.19/1.77	2.45	0.016	6.09 *	0.05
								Total	0.28
SD	SD baseline	0.27	0.09	0.27	0.09/0.45	2.93	0.004	20.42 ***	0.16
	Educational level	1.00	0.34	0.26	0.34/1.67	3.00	0.003	7.96 **	0.06
	Community integration baseline	0.92	0.44	0.20	0.05/1.79	2.10	0.039	6.32 *	0.04
	Resilience 12 m	1.03	0.47	0.20	0.10/1.96	2.19	0.031	4.79 *	0.03
								Total	0.29
SI	SI baseline	0.32	0.07	0.35	0.18/0.47	4.40	<0.001	23.80 ***	0.16
	Comorbidity	−1.21	0.58	−0.17	−2.36/−0.06	−2.09	0.039	6.62 *	0.05
	Resilience 12 m	1.74	0.43	0.33	0.90/2.58	4.09	<0.001	16.72 ***	0.10
								Total	0.31
RI	Satisfaction with social support baseline	1.89	0.53	0.47	0.83/2.95	3.60	0.001	16.14 ***	0.27
	Community integration baseline	1.22	0.58	0.28	0.06/2.39	2.12	0.041	4.49 *	0.06
								Total	0.33
Total QoL Index	Total baseline	0.55	.11	0.53	0.33/0.76	5.00	<0.001	25.03 ***	0.17
	Dependency level	−8.27	2.01	−0.39	−12.27/−4.27	−4.11	<0.001	15.61 ***	0.15
	Depression baseline	−5.63	2.26	−0.24	−10.14/−1.13	−2.49	0.015	8.12 **	0.07
	Satisfaction with social support baseline	4.50	2.04	0.21	0.43/8.57	2.20	0.031	4.84 *	0.04
								Total	0.43

Note. EW = emotional well-being; IR = interpersonal relationships; MW = material well-being; PD = personal development; PW = physical well-being; SD = self-determination; SI = social inclusion; RI = rights; * *p* < 0.05, ** *p* < 0.001, *** *p* < 0.0001.

## Data Availability

The data presented in this study are available on request from the corresponding author. The data are not publicly available due to the ethical demands.

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
