# Peer review of "Predictive Factors of Self-Reported Quality of Life in Acquired Brain Injury: One-Year Follow-Up"

_ijerph, 2021, doi:10.3390/ijerph18030927_

Round 1
Reviewer 1 Report
Thank you for the opportunity to see this paper.
I found the paper focused on an important topic. The expansion of the concept of QoL is important to explore not only in this population but in others.
I found the paper to be well written. The ideas were logically presented and clear. I see how the background literature summary provided a basis for the study. The research objectives were clearly stated. The design and analysis was appropriate for the objectives. The analysis was described in detail. The results are clearly presented in the text and the tables. The discussion is based on the results.
I only have two comments:
1) I think it would be helpful for an international audience to know who actually goes to the rehabilitation centres from which you accrued participants. In some countries, these centres may not exist or are not for all patients.
2) implications for practice and future research are not really included. Only one sentence at the end really focuses clearly on this. Could there be some elaboration?
Author Response
Reviewer # 1
Dear Reviewer # 1
First, we would like to show our gratitude with you for all your efforts and the time you invested in reviewing the manuscript entitled “Predictive factors of self-reported quality of life in acquired brain injury: one-year follow-up” (reference ijerph-1037672) that we submitted to the International Journal of Environmental Research and Public Health. We also appreciate your positive assessment of our work and we have carefully considered your comments and suggestions, which have undoubtedly contributed to a significant improvement in our work.
This letter is structured as follows: We copy and highlight in bold each comment and suggestion, and below we answer to each of them showing both the rationale underlying our answer, as well as the place in the manuscript (indicating section, page, and lines) where we have added/modified/deleted what you suggested. All changes added to the text are marked in red in the manuscript, so that you can check all the changes expeditiously.
Once again, thank you for your time.
Best regards,
The authors
Reviewer # 1 Comments to Author
Thank you for the opportunity to see this paper.
I found the paper focused on an important topic. The expansion of the concept of QoL is important to explore not only in this population but in others.
I found the paper to be well written. The ideas were logically presented and clear. I see how the background literature summary provided a basis for the study. The research objectives were clearly stated. The design and analysis was appropriate for the objectives. The analysis was described in detail. The results are clearly presented in the text and the tables. The discussion is based on the results.
Comment # 1. I think it would be helpful for an international audience to know who actually goes to the rehabilitation centres from which you accrued participants. In some countries, these centres may not exist or are not for all patients.
We totally agree with this comment. The rehabilitation system in Spain is not the same as in other countries, even within Spain there are profound differences between the different regions. Following your suggestion, information has been added on the differences between rehabilitation and day centers, to which the sample participating in this study attended (see Materials and Methods, procedure section, page 4, lines 157-165): “Participants attended to ABI-specific care centres (i.e., rehabilitation centres and day centres) spreaded throughout Spain. These are socio-sanitary centres focused mainly on a chronic phase of ABI (although when the rehabilitation phase within hospitals is brief due to time or resources constrain, people with ABI in the subacute phase are can be also sent to these centres). The main difference between rehabilitation centres and day care centres is the purpose of the services provided: day care centres provide daily care services with the aim of improving or maintaining the best possible level of personal autonomy and provide family respite for caregivers. The rehabilitation centres pursue a therapeutic objective, aimed at re-educating and compensating for the consequences of the injury, preventing future complications, and improving the preserved abilities.”
Comment # 2 Implications for practice and future research are not really included. Only one sentence at the end really focuses clearly on this. Could there be some elaboration?
Once again, we absolutely agree with the reviewer. The conclusions section, although it should be brief, is too short, especially regarding practical implications and future research. For this reason, the section has been reformulated, adding the following information (Conclusions, page 15, lines 386-392): “This implies the need to implement a greater number of programmes and early actions on these aspects for clinical practice, being especially important the prevention and detection programs. Furthermore, these results highlight the importance of psychological, neuropsychological, and occupational therapy intervention within the care of ABI, which has been limited in many cases to physical aspects. Future lines of work should include broader longitudinal follow-ups, as well as analysis of the specific effects that some of the interventions carried out in this population have on QoL.”
Reviewer 2 Report
Thank you for the opportunity to review the paper Predictive factors of self-reported quality of life in acquired brain injury: One year follow up. This paper is extremely well written however there are a number of issues with the analysis that need to be addressed.
Main methodological issues
- QoL as an outcome: My main concern for the study is that the analysis is not for quality of life (QoL) at 1-year post ABI, but 1-year post entry into the study. This was quite difficult to discern in the current manuscript and should be made clearer. In addition, the analyses need to be changed to take into account time since ABI.
To answer the question ‘What factors predict QoL in participants with ABI’: This analysis should examine the Time 1 sociodemographic and clinical characteristics associated with QoL at entry to the study. All analyses should control for time since injury.
- Type of centre impact on results: Participants were recruited through both day centres and rehabilitation centres. Although I am not familiar with the medical system in Spain, I would expect that rehabilitation centres would be used early in the patient recovery journey before the full extent of the ABI and its impact is known and would have a focus on recovery and rehabilitation. On the other hand, day centres would be more likely to be used in the long term for occupational therapy and respite for carers. Therefor, the long term impact of ABI may not yet be fully known and realised in participants recruited through rehabilitation centres. Therefor, when examining QoL, all analyses also need to control for type of centre.
- The second interesting question that can be answered by this data, is what predicts changes in QoL in people with ABI. Table 1 shows that the mean time since injury was 8.25 years (SD 7.83; range 0.5-47.5 years). The authors need to identify if QoL has improved for any participant, and then examine what are the predictors of improvement in QoL (or worsening, as the case may be).
Abstract
Note that participants were people who had experienced an ABI from 0.5-47.5 years ago.
See main methodological issues above.
Did families complete any of the instruments as proxy for the person with ABI?
Introduction
Please provide some more information about the prevalence, causes, and severity, of acquired brain injury (ABI) in the community and the long term impact of ABI.
Lines 78-84: This study does not focus on the predictive factors of QoL over a one-year follow up (see main methodological concerns above). This paragraph needs revising.
Materials and Methods
Study participants: Please include a participant recruitment and flow chart to determine how may participants were screened, the outcome of screening, and those who refused to participate; need clear description of attrition and any significant differences in those who dropped out and those who remained in the study at 12 months;
In the section on Procedure, you report that consent was obtained from participants and their relatives and you briefly review the report by proxy in your Introduction. However, you do not describe if or when data was provided by the relatives in proxy for the person with ABI. Please ensure this is clearly described;
Instruments: Use past tense to describe instruments; include reliability and validity information for your study; describe more clearly what subscales were calculated for each instrument and included in the analyses; Patient Competency rating Scale: you report the responses from participants with an ABI were compared to professionals. What does this mean?
Procedure: See main methodological concerns above.
Minor editing
Page 1
Line 18: replace ‘had had’ with ‘had experienced’
Line 19: reference required.
Line 21: When were the assessments carried out?
Line 38: remove ‘However, traditionally, the’
Line 41: Remove ‘the following’
Page 2
Line 58: Remove ‘It sems that the QoL’s’ to read “Improvement in QoL are generally experienced up to’
Line 68: New paragraph from ‘Regarding the published…’
Line 71: Are impairments the same as comorbidities?; remove ‘In any case’
Line 74: Insert ‘good’ social support
Materials and Methods
Study participants: Use past tense
Line 92: Exclusion criteria was (c) not being able to understand or answer most questions: Who was this assessed by and how did you assess this.
Results
Line 170: p-values in Table 1 not required.
Line 171-173: review
Results and Discussion will need to be reviewed in consideration of above concerns.
Author Response
Reviewer # 2
Dear Reviewer # 2
First, we would like to show our gratitude with you for all your efforts and the time you invested in reviewing the manuscript entitled “Predictive factors of self-reported quality of life in acquired brain injury: one-year follow-up” (reference ijerph-1037672) that we submitted to the International Journal of Environmental Research and Public Health. Without any doubt, your comments and suggestions have been very helpful to improve the manuscript, and we have carefully considered all of them.
This letter is structured as follows: We copy and highlight in bold each comment and suggestion, and below we answer to each of them showing both the rationale underlying our answer, as well as the place in the manuscript (indicating section, page, and lines) where we have added/modified/deleted what you suggested. All changes added to the text are marked in red in the manuscript, so that you can check all the changes expeditiously.
Once again, thank you for your time.
Best regards,
The authors
Reviewer # 2 Comments to Author
Thank you for the opportunity to review the paper Predictive factors of self-reported quality of life in acquired brain injury: One year follow up. This paper is extremely well written however there are a number of issues with the analysis that need to be addressed.
Main methodological issues
Comment # 1. QoL as an outcome: My main concern for the study is that the analysis is not for quality of life (QoL) at 1-year post ABI, but 1-year post entry into the study. This was quite difficult to discern in the current manuscript and should be made clearer. In addition, the analyses need to be changed to take into account time since ABI.
To answer the question ‘What factors predict QoL in participants with ABI’: This analysis should examine the Time 1 sociodemographic and clinical characteristics associated with QoL at entry to the study. All analyses should control for time since injury.
Thank you for this suggestion as it will help us to better focus the article. Indeed, the interpretation you make is adequate, the one-year follow-up is since the inclusion in the study and not since the ABI occurred. First of all, and to make this matter clearer to the reader, some small changes have been introduced:
- Abstract: “The sample comprised 203 adults with ABIs (64% male) aged 18-86 years (M = 53.01, SD = 14.44). Stroke was the main etiology of the injury (55.7%), followed by a TBI (32.8%), and the average time since injury was 8 years (M = 25, SD = 7.83, range = 0.5–47.5)” (page 1, lines 17-19) and “The levels of dependency, depression, and satisfaction with social support were independent predictors of the total QoL score one-year follow-up” (page 1, lines 27 and 28).
- Introduction (page 2, lines 90-92): “In the present manuscript, we are going to focus on predictive factors of QoL over a one-year follow-up from baseline, using a multidimensional model of the concept and self-report evaluations.”
- Results (Patient sample section, page 5, lines 217-219): “Stroke was the main aetiology of the injury (55.7%), followed by a TBI (32.8%), and the average time since injury was 8 years (M = 25, SD = 7.83, range = 0.5–47.5).”
- Discussion (page 13, lines 297 and 298): “This study presents the QoL outcomes, as measured by the CAVIDACE scale, at the one-year follow-up in a sample of ABI adults whom had experienced ABI some years ago.”
Second, we agree with the reviewer that there is a great temporal fan in the time elapsed since ABI. However, as has been reported in the documents cited in the manuscript (Introduction, page 2, lines 64-69), after a temporary window of 2 or 3 years, a chronic phase is entered and few changes in QoL occurs. For greater clarity in the presentation of this letter, this issue has continued to be addressed in the response to your comment 3.
Finally, we would like to finish addressing the last aspect mentioned by the reviewer in this comment about the need to use QoL in the baseline as dependent factor to build the predictive model. The use of predictive models in which evaluations of the aspects that constitute the dependent and independent variables are taken at the same time is common in the literature. However, to our understanding, these models would be explanatory rather than predictive, since the prediction requires that the problems that are occurring at a given time (in this case, the baseline assessment) can predict another variable (in this case, QoL) at a future time (in this case, a year later). We hope we have addressed this issue properly. However, we are willing to modify it if the reviewers and the editor consider it appropriate.
For further clarification of this point and to verify its practical application in a scientific article in the field of QoL in people with ABI, we refer the reviewer to reference 10 of the manuscript:
Forslund, M. V.; Roe, C.; Sigurdardottir, S.; Andelic, N. Predicting Health-Related Quality of Life 2 Years after Moderate-to-Severe Traumatic Brain Injury. Acta Neurol. Scand., 2013, 128 (4), 220–227. https://doi.org/10.1111/ane.12130.
Furthermore, in the predictive models constructed in our study the QoL’s levels at the baseline were entered as independent variables in a first step, in order to isolate this effect from the rest of the variables of interest.
Comment # 2. Type of centre impact on results: Participants were recruited through both day centres and rehabilitation centres. Although I am not familiar with the medical system in Spain, I would expect that rehabilitation centres would be used early in the patient recovery journey before the full extent of the ABI and its impact is known and would have a focus on recovery and rehabilitation. On the other hand, day centres would be more likely to be used in the long term for occupational therapy and respite for carers. Therefor, the long term impact of ABI may not yet be fully known and realised in participants recruited through rehabilitation centres. Therefor, when examining QoL, all analyses also need to control for type of centre.
The response to this comment was previously addressed (comment 1 of reviewer 1), about the types of recruited centres attended by the participants in our study. It should be noted that the rehabilitation centres mentioned are sociosanitary centres, resources usually destinated for people with ABI in our country during the chronic phase (and different from hospital rehabilitation resources, which are limited to the first 3 months after the injury, approximately. The research team had no contact with users of these types of resources).
However, based on your consideration, we believe that it is necessary to briefly clarify in the text the terms of day centre and rehabilitation centre for an international audience that is not familiar with the Spanish system (see Materials and Methods, procedure section, page 4, lines 157-165): “Participants attended to ABI-specific care centres (i.e., rehabilitation centres and day centres) spreaded throughout Spain. These are socio-sanitary centres focused mainly in a chronic phase of ABI (although when the rehabilitation phase within hospitals is brief due to time or resources constrain, people with ABI in the subacute phase are can be also sent to these centres). The main difference between rehabilitation centres and day care centres is the purpose of the services provided: day care centres provide daily care services with the aim of improving or maintaining the best possible level of personal autonomy and provide family respite for caregivers. The rehabilitation centres pursue a therapeutic objective, aimed at re-educating and compensating for the consequences of the injury, preventing future complications, and improving the preserved abilities.”
Regarding the reviewer's concern about the absence of deficits’ settlement in the people who receive care in the rehabilitation centres, after having clarified that the time elapsed since ABI does not always have a direct relationship with the type of resource and also taking into account that in the earliest cases, 6 months or more had elapsed since the ABI (and therefore the sequelae are already quite stabilized), we believe that it is not necessary to control the variable "care resource" in the analyzes. However, we are willing to do it if the reviewers and the editor consider it appropriate.
Comment # 3. The second interesting question that can be answered by this data, is what predicts changes in QoL in people with ABI. Table 1 shows that the mean time since injury was 8.25 years (SD 7.83; range 0.5-47.5 years). The authors need to identify if QoL has improved for any participant, and then examine what are the predictors of improvement in QoL (or worsening, as the case may be).
In relation to this comment, we would like to point out that this study is part of a larger and more exhaustive study, which includes some of these results. This is mentioned in the introduction part (Introduction, page 2, line 86). Some of the results can be consulted in the following two publications that we indicate below.
Aza, A.; Verdugo, M. Á.; Orgaz, M. B.; Fernández, M.; Amor, A. M. Adaptation and Validation of the Self-Report Version of the Scale for Measuring Quality of Life in People with Acquired Brain Injury (CAVIDACE). Qual. Life Res., 2020, 29 (4), 1107–1121. https://doi.org/10.1007/s11136-019-02386-4.
Aza, A.; Verdugo, M. Á.; Orgaz, M. B.; Andelic, N.; Fernández, M.; Forslund, M. V. The Predictors of Proxy- and Self-Reported Quality of Life among Individuals with Acquired Brain Injury. Disabil. Rehabil., 2020. https://doi.org/10.1080/09638288.2020.1803426.
In addition, one more paper has recently been accepted by the International Journal of Clinical and Health Psychology (new reference 42):
Verdugo, M. A.; Aza, A.; Orgaz, M. B.; Fernández, M.; Amor, A. M. Longitudinal study of quality of life in acquired brain injury: a self- and proxy-report evaluation. Int. J. Clin. Health Psychol.
Specifically, this latest publication addresses the issue mentioned by the reviewer in this comment and that is what are the changes that occur in QoL’s levels in people with ABI during this year of follow-up. In order to facilitate the reader's understanding of this paper and to be able to complete the information that we report, it has been added the following information (Introduction, pages 2 and 3, lines 86-96): “This manuscript is based on an extensive research about QoL after an ABI [40, 41]. In another manuscript [42], significant changes have been found in QoL between baseline and one-year follow-up in almost all domains (emotional well-being, material well-being, personal development, physical well-being, and rights) and total QoL, being the respondents both the person with ABI (self-report) and the professionals and relatives (proxy-report). In the present manuscript, we are going to focus on predictive factors of QoL over a one-year follow-up from baseline, using a multidimensional model of the concept and self-report evaluations. We aimed to (1) describe changes in QoL after an ABI between baseline and one-year follow-up evaluation, (2) describe and explore the changes in important variables (i.e., depression, self-awareness, community integration, resilience, and social support) at the one-year follow-up, and (3) examine the impact of sociodemographic, injury-related, personal, and social variables on QoL and identify the predictors of a better QoL.”
An attachment has been added with the aforementioned manuscript so that the reviewer can have more exhaustive access to the aforementioned information.
In addition to the information about this article, the data mentioned by the reviewer have been included in this manuscript to analyze the changes in the QoL’s domains and total score (self-report evaluations) between the baseline and one-year follow-up, but introducing how the variable “time since ABI” affects to the results.
The time that has elapsed since the ABI has been an issue that worried the reviewer and we would like to return to this comment. As we mentioned earlier, there is a time after the ABI of two or three years which has normally been reported in the literature as the window time in which the greatest changes in QoL levels usually occur.
In the reference 42 mentioned above, you could see that the authors analyzed whether the QoL experienced different patterns of change whether the patients had a recent ABI (i.e., 3 years ago or less) or were chronic patients (i.e., ABI more than 3 years ago); using the proxy evaluation of the professionals and also including an intermediate follow-up that was carried out at six months. In that case, the time since ABI was found to be a significant factor. On this occasion we have repeated a similar process with the evaluation of the self-report (so these results had not been published in any other manuscript).
First, the first goal of the manuscript has been slightly modified to include this question (see Introduction, pages 2 and 3, lines 92 and 93): “(1) describe changes in QoL after an ABI between baseline and one-year follow-up evaluation”.
Second, information on the new analyzes carried out has been added in the statistical analysis section (see Material and Methods, page 5, lines 191-194): “To verify the effect that the time elapsed since ABI could have on the changes experienced in the QoL, the sample was divided between those who had recently had the ABI (i.e., 3 years ago or less) and those who were in a chronic phase (i.e., more than 3 years ago) and carried out repeated measured t-test between QoL at baseline and one-year follow-up for QoL’s domains and total scores.”
Third, the corresponding data has been added in the results section, including a new head called (see page 7, lines 234- 245) "3.2. Changes in QoL from baseline to one-year follow-up."
A table (Table 3) and the following information have been added to this section: “Improvements in QoL levels have been reported between baseline and one-year follow-up (see Table 3 for more information). However, these changes have only been significant in emotional well-being for the most recent ABI group (i.e., ABI 3 years ago or less) and in personal development for the chronic group (i.e., ABI more than 3 years ago).”
Based on these analyzes, we have been able to verify that the time elapsed since ABI does not exert a significant effect on the changes experienced in the QoL between baseline and the one-year follow-up, as it happened in the evaluation carried out by professionals (proxy-report evaluation) that was reported in the aforementioned reference 42. In conclusion, with this information and considering that the time elapsed since ABI was not a significant variable in the QoL’s levels for any of the domains or the total score (see Table 3, now Table 4), it has been decided not to include this variable in the rest of the analyzes carried out in the manuscript.
Abstract
Comment # 4. Note that participants were people who had experienced an ABI from 0.5-47.5 years ago.
This aspect was addressed in comment 1 by adding some clarification on the follow-up time and the time elapsed since ABI. One of them was in the abstract (page 1, line 17-19): “The sample comprised 203 adults with ABIs (64% male) aged 18-86 years (M = 53.01, SD = 14.44). Stroke was the main aetiology of the injury (55.7%), followed by a TBI (32.8%), and the average time since injury was 8 years (M = 8.25, SD = 7.83, range = 0.5–47.5).”
Comment # 5. See main methodological issues above.
Once the explanations have been presented in the previous comments and the appropriate changes have been made, no further modifications have been included in the abstract. However, we are willing to do it if the reviewers and the editor consider it appropriate.
Comment # 6. Did families complete any of the instruments as proxy for the person with ABI?
This study is part of a more extensive investigation in which the proxy version of the CAVIDACE scale was used with professionals and relatives. However, we limit this manuscript to present only the self-report data. This issue has also been addressed in comment 10.
Introduction
Comment # 7. Please provide some more information about the prevalence, causes, and severity, of acquired brain injury (ABI) in the community and the long term impact of ABI.
The first paragraph of the introduction has been modified to add the information mentioned (see page 1, lines 36-44): “Acquired brain injuries (ABIs) is caused by a sudden injury to the brain that occurs after birth and includes different diagnoses such as traumatic brain injury (TBI), stroke, brain tumor, anoxia, and infection. In Spain, there is a prevalence of 420,064 people with ABI and approximately 104,701 new cases per year [1], imposing considerable costs on society due to the years of life lost to disability or death [2]. ABI is often accompanied by long-lasting or permanent physical (spasticity, mobility problems and chronic pain), cognitive (executive functioning, attention, memory and learning, communication and anosognosia), emotional (anxiety and depression), and social impairments (social isolation and inability to return to work) [3–8] what it entails a diminished quality of life (QoL) [8–12].”
The reference number 1 ([1]) has been added: Quezada, M. Y.; Huete, A.; Bascones, L. M. Las Personas Con Daño Cerebral Adquirido En España; FEDACE y Ministerio de Salud, Seguridad Social e Igualdad: Madrid, 2015.
Comment # 8. Lines 78-84: This study does not focus on the predictive factors of QoL over a one-year follow up (see main methodological concerns above). This paragraph needs revising.
We appreciate this comment aimed at clarifying the characteristics of the study. However, a question arises in this regard: we do not know if the concern is directed to the confusion that could exist regarding the time elapsed since the injury with the follow-up time (we believe that we have clarified this aspect throughout the document) or if it refers to the need to introduce QoL levels in the baseline. This last aspect has been extensively addressed in the last part of comment 1.
This paragraph has been reformulated (see Introduction, pages 2 and 3, lines 86-96): “This manuscript is based on an extensive research about QoL after an ABI [40, 41]. In another manuscript [42], significant changes have been found in QoL between baseline and one-year follow-up in almost all domains (emotional well-being, material well-being, personal development, physical well-being, and rights) and total QoL, being the respondents both the person with ABI (self-report) and the professionals and relatives (proxy-report). In the present manuscript, we are going to focus on predictive factors of QoL over a one-year follow-up from baseline, using a multidimensional model of the concept and self-report evaluations. We aimed to (1) describe changes in QoL after an ABI between baseline and one-year follow-up evaluation, (2) describe and explore the changes in important variables (i.e., depression, self-awareness, community integration, resilience, and social support) at the one-year follow-up, and (3) examine the impact of sociodemographic, injury-related, personal, and social variables on QoL and identify the predictors of a better QoL.”
Materials and Methods
Comment # 9. Study participants: Please include a participant recruitment and flow chart to determine how may participants were screened, the outcome of screening, and those who refused to participate; need clear description of attrition and any significant differences in those who dropped out and those who remained in the study at 12 months;
We greatly appreciate your suggestion. We have tried to reflect as clearly as possible the participant recruitment process, and we are very sorry if it has not been approached appropriately. We believe that now it is difficult to include the information as you request it, since at the time of recruitment of the sample the research team contacted the care centres directly. For this reason, we cannot represent the flow chart. Nevertheless, to overcome this limitation, information about the participant recruitment process has been added in the procedure section (see Materials and methods, page 4, lines 150-156): “Participating organizations that provide attention to the ABI population were recruited through emails and telephone calls. First, we contacted the centres that had participated in the study in which the CAVIDACE scale was constructed and validated. On numerous occasions, these professionals facilitated liaisons with other centres, thereby resulting in snowball sampling. Second, in order to recruit a larger sample, information about the study was (a) disseminated through conferences and (b) posted on the university’s website. Of the 32 centres with which he contacted, 26 of them finally agreed to participate in the study.”
And (lines 166-170) “Once a centre expressed interest in participating, a research team member visited it and provided all necessary information about the study. In each centre, a research assistant was trained to oversee the administration of the instruments. To participate in the study, participants had to meet the established inclusion criteria, and when the number of participants exceeded the possibilities of participation of the centre, it was the research assistant who randomly selected participants.”
Moreover, we can provide detailed information on those participants who started the study and those who dropped out at 12 months. In this sense, several changes have been introduced:
- In Table 1 all the descriptive data that had already been reported for the initial sample in the study have been added. In addition, the corresponding statistics have been calculated to determine the differences between both samples. These results have been presented in Table 1.
- In the statistical analysis section (see Materials and Methods, page 5, line 187-190), the information referring to these new added analyzes has been added: “Descriptive data are displayed as the mean, SD and range or absolute and relative frequencies. When comparing characteristics between the included patients and those who were lost to follow-up, the categorical variables were analyzed with chi-squared test and the continuous variables with independent t-test.”
- As well as the main results in the results section (see page 5, line 219-221): “When comparing the differences between patients with and without follow-ups, significant differences were found in time since injury (t371 = -95, p<.01), etiology (χ² = 24.30, p<.001), and type of centre (χ² = 23.36, p<.001).”
Comment # 10. In the section on Procedure, you report that consent was obtained from participants and their relatives and you briefly review the report by proxy in your Introduction. However, you do not describe if or when data was provided by the relatives in proxy for the person with ABI. Please ensure this is clearly described;
Thank you, this is a very important comment that has made us see that we were generating confusion about it. Although this manuscript is part of a more extensive investigation, which also includes, in addition to the self-report evaluation, proxy evaluations of relatives and professionals, in this manuscript we focused exclusively on the self-evaluations.
Therefore, to avoid possible confusion, it has been eliminated from this manuscript that the informed consent of the family was obtained (see Materials and Methods, procedures section, page 5, line 179). Likewise, we have kept the brief reference that is made in the introduction about proxy instruments, since we want the reader to bear in mind this double aspect of QoL’s assessment.
Comment # 11. Instruments: Use past tense to describe instruments; include reliability and validity information for your study; describe more clearly what subscales were calculated for each instrument and included in the analyses; Patient Competency rating Scale: you report the responses from participants with an ABI were compared to professionals. What does this mean?
Thanks for this suggestion. We have corrected the verb tense used in the description of the instruments, using past tenses (see pages 3 and 4).
Likewise, information on the reliability and validity of our study has been added (see Materials and Methods, Measures section, CAVIDACE scale, page 3, lines 121-124): “Its psychometric properties were good and comparable to those of the original scale: QoL is composed of eight first-order intercorrelated domains (CFI = 0.891, RMSEA = 0.050, TLI = 0.881), and the internal consistency was adequate in seven of the eight domains (ω = 0.66.–0.87) [39].”
Regarding the third question, we would like to indicate that we worked with domains only on some scales (specifically, with the QoL scale, CAVIDACE, and with the social support scale, SSQ6). Therefore, information on these two scales has been added:
- CAVIDACE scale (see Materials and Methods, Measures section, page 3, lines 112-121): “This version consisted of 40 items, which assessed the eight domains that are subsumed by Schalock and Verdugo’s model: emotional well-being, interpersonal relationships, material well-being, personal development, physical well-being, self-determination, social inclusion, and rights. The responses were recorded on a four-point scale: 0 = never, 1 = sometimes, 2 = frequently, and 3 = always. The instrument includes negatively worded items, which were reversed before summing the scores for each domain. These direct scores are transformed into standard scores for each domain (M = 10, SD = 3) and percentiles. Moreover, the scale provides an over-all raw QoL score (i.e., the sum of the direct scores obtained in each of the domains) that may vary from 0 to 120, where higher scores indicate higher QoL. This overall score may be converted into an easily interpretable QoL Global Index (M = 100; SD = 15).”
- Social Support Questionnarie-6 (see Materials and Methods, Measures section, page 4, lines 143-148).
The information appeared like this: “Individuals are required to respond to the 6 items indicating the number of available supports and their level of satisfaction. Scores can range from 0 to 6 and 1 to 6, respectively.”
And it has been reformulated as follows: “Individuals were required to respond the 6 items by (a) indicating the number of individuals who are available to provide them with support and (b) rate their level of social support satisfaction. Scores can range from 0 (no social support) to 6 (very high social support) for the number of supports, and 1 (very unsatisfied) to 6 (very satisfied) for the satisfaction domain in each item or area. From these scores in the 6 areas, an average score was calculated for the number of supports and for satisfaction.”
In the rest of the instruments we worked with the global score, calculated from the sum of scores, as specified in the scientific articles and manuals of the aforementioned tests.
There is an exception, and it is precisely the last issue that concerned the reviewer in this comment, and it is the correction system of the anosognosia scale, Patient Competency Rating Scale (see Materials and Methods, Measures section, page 3, lines 129-134). This system is based on the calculation of the level of anosognosia from the differences found between the person with ABI and an objective external evaluator on the patient's abilities to perform certain daily living tasks. The information appeared like this: “The PCRS consists of 30 items that assesses the competency to perform different daily living tasks [44]. The responses of those with ABIs were compared to professionals. Wider discrepancies indicated poor self-awareness.” And it has been reformulated as follows: “The PCRS consisted of 30 items that assessed the competency to perform different daily living tasks [44]. Individuals with ABI are required to indicate the extent to which it is difficult for them to perform the task that is described in each item. Their responses can be compared to those of a family member or professional to determine the self-awareness’ level. In this study, we compared the responses of individuals with ABI and professionals. Wider discrepancies indicated poor self-awareness.”
Another question to take into account about the scale score calculation systems is the system used by the research team to categorize these quantitative variables in order to perform both the ANOVA and the regression tests (tables 3 and 4, respectively). In this sense, a categorization of the variables was carried out in levels: low, intermediate and high, based on the percentile calculation. Information about this has been included in the statistical analysis section (see Materials and Methods, page 5, lines 197-200): “Analysis to determine the predictors of the QoL scores was conducted. Before carrying out the analyzes, we implemented a transformation of the quantitative scales (i.e., PHQ9, PCRS, CIQ, CD-RISC, and SSQ6) in categories (i.e., low, intermediate, and high) from the calculation of the percentiles.”
Comment # 12. Procedure: See main methodological concerns above.
Significant changes have been introduced in the procedure section, which have been reported in the previous comments.
Minor editing
Comment # 13. Page 1. Line 18: replace ‘had had’ with ‘had experienced’.
Thank you for this comment. We have modified the term as suggested.
Comment # 14. Page 1. Line 19: reference required.
The reference of the self-report version of the CAVIDACE scale, that was used in this study, has been added: Aza, A.; Verdugo, M. Á.; Orgaz, M. B.; Fernández, M.; Amor, A. M. Adaptation and Validation of the Self-Report Version of the Scale for Measuring Quality of Life in People with Acquired Brain Injury (CAVIDACE). Qual. Life Res., 2020, 29 (4), 1107–1121. https://doi.org/10.1007/s11136-019-02386-4.
Comment # 15. Page 1. Line 21: When were the assessments carried out?
In the study presented, both the CAVIDACE scale (self-report version) and the complementary tests were applied at the baseline and at the one-year follow-up. In the case of sociodemographic and clinical variables, they were only collected at the baseline. This information was compiled in the Materials and Methods (see procedure section, line 175-177).
The following clarification has also been added in the abstract (Page 1, lines 20-22): “Other variables measured were: depression, self-awareness, community integration, resilience, and social support at baseline and one-year follow-up.”
Comment # 16. Page 1. Line 38: remove ‘However, traditionally, the’
Thank you for the comment. The phrase has been removed following the reviewer's suggestion
Comment # 17. Page 1. Line 41: Remove ‘the following’
Thank you for the comment. The phrase has been removed following the reviewer's suggestion
Comment # 18. Page 2. Line 58: Remove ‘It sems that the QoL’s’ to read “Improvement in QoL are generally experienced up to’
Thank you for the comment. The correction has been made: “Improvement in QoL are generally experienced up to one [5, 7, 21–23, 25] or two [11, 24] years after an injury, and, afterwards, the levels remain more or less stable [3, 6, 9].”
Comment # 19. Page 2. Line 68: New paragraph from ‘Regarding the published…’
Thank you for the comment. The paragraph has been separated following the reviewer's suggestion.
Comment # 20. Page 2. Line 71: Are impairments the same as comorbidities?; remove ‘In any case’
Indeed, when we speak of comorbidity we mean the presence of several impairments in the same person and that the QoL worsens as the number of these impairment increases. The phrase has been reformulated to improve its understanding (Introduction, page 2, line 79 and 80): “However, there is an agreement that a greater number of impairments after an ABI (i.e., comorbidity) are related to a worse QoL [2, 16, 21, 30, 33, 34].”
The phrase “In any case” has been removed following the reviewer's suggestion.
Comment # 21. Page 2. Line 74: Insert ‘good’ social support
Thank you for the comment. The correction has been made in the terms suggested by the reviewer.
Comment # 22. Materials and Methods. Study participants: Use past tense
Thank you for the comment. The correction has been made (see Materials and Methods, study participants section, line 100-104): “The ABI participants had to meet the following inclusion criteria: (a) had an ABI, (b) were at least 16 years, (c) were treated in an ABI-specific centre, and (d) had signed an informed consent form. The exclusion criteria were the following: (a) were in a state of coma or having minimum consciousness, (b) had global aphasia, and (c) not were able to understand or answer most questions.”
Comment # 23. Materials and Methods. Line 92: Exclusion criteria was (c) not being able to understand or answer most questions: Who was this assessed by and how did you assess this.
The professional of each centre was in charge of determining the person's ability to respond to the test, following the guidelines and indications of the research team. So, information about it has been added in the procedure section (see page 5, lines 166-174): “Once a centre expressed interest in participating, a research team member visited it and provided all necessary information about the study. In each centre, a research assistant was trained to oversee the administration of the instruments. To participate in the study, participants had to meet the established inclusion criteria, and when the number of participants exceeded the possibilities of participation of the centre, it was the research assistant who randomly selected participants. In addition, the research assistant (in consensus with a professional from the centre, when deemed necessary) was in charge of determining the ability of the person with ABI to answer the questionnaires. The research team provided printed copies, although respondents were able to complete the scales online as well.”
However, we are aware that it constitutes a limitation, despite the fact that professionals are an expert source with extensive knowledge of patients and their impairments, and therefore it was included as such in the original manuscript (see discussion, page 14, lines 375-378): “Second, no objective test was used to determine the ability of people with ABI to participate in the study. This could introduce some subjectivity, even though the research team strictly controlled it and the professionals acted on their clinical judgement based on their knowledge of the individuals with ABIs and their clinical history.”
Comment # 24. Results. Line 170: p-values in Table 1 not required
Sorry, this is a typo. However, this information has been kept due to the new information added in this table, as reported in the comment 9.
Comment # 25. Results. Line 171-173: review
Thank you for the comment. Table 1 has been reformulated to add the sociodemographic and clinical data of the sample that did not complete the follow-up evaluation and the differences of this with the study sample.
Comment # 26. Results. Results and Discussion will need to be reviewed in consideration of above concerns.line
According to this comment, a new section (see “3.2. Changes in QoL from baseline to one-year follow-up”) has been added in the results that includes a new table (Table 3).
This new information has also been briefly collected in the discussion (see page 13, lines 309-315): “Between the baseline evaluation and the one-year follow-up, improvements in QoL have generally been experienced, but they were significant only in emotional well-being and personal development. In other studies [42] that analyzed different change patterns as a function of the time elapsed since the ABI, more significant changes were obtained when ABI was recent; however, these results have not been replicated in the present study. The determining factor may be that in this case we are faced with self-report evaluations compared to the proxy evaluations used in other cases.”